# Multi-Criteria Analysis of the Mass Tourism Management Model Related to the Impact on the Local Community in Constanta City (Romania)

**Catalin Anton \*, Angela-Eliza Micu and Eugen Rusu** 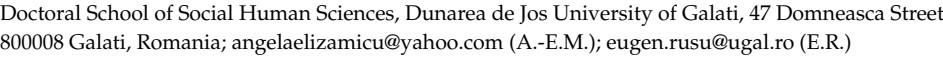

Doctoral School of Social Human Sciences, Dunarea de Jos University of Galati, 47 Domneasca Street, 800008 Galati, Romania; angelaelizamicu@yahoo.com (A.-E.M.); eugen.rusu@ugal.ro (E.R.)
\* Correspondence: catanton@gmail.com

**Abstract:** Traditionally and socially, the tourism in Constanta is considered to be important to the local economy. Sun and beach locations are both a draw for locals and tourists to the city, on the Black Sea. However, vacation-oriented activities in the city only have a seasonal cycle. In this paper, we proposed to analyze the mass tourist activity in Constanta, taking into account economic, social, and environmental conditions. Additionally, we attempted to build a model based on the data available. The model was developed using a PESTEL analysis to determine the supportability factor of the indicators identified. We also set out to create a projection of the activities proposed for analysis by 2050. To create a model for coastal areas, the data used in this research must be accurate and consistent. Furthermore, correctly identifying indicators and their relationships is a critical step in conducting a thorough study. Last but not least, finding the calculation coefficient for the activity in question is critical, as collecting data from various activities might be challenging when trying to find a feasible model.

**Keywords:** mass tourism; management model; Constanta; PESTEL; Black Sea; sustainability





## 1. Introduction

There is interdisciplinarity in regards to the science that has been carried out in coastal areas. Due to the interdependence of land and sea, it brings several threats to biodiversity and quality of life in this coastal area [1]. Understanding both the natural and the systems processes that keep them in harmony, is a means to understand the complementary nature of their relationship [2].

Integrative application of pressure appears to be the optimal solution for the coastal zone's longer-term growth.

It is particularly significant for the European Union, given that it includes areas of over 68,000 km of coastline with an integrated approach to management. Even with these significant disadvantages and weaknesses, the administration looks to be simpler with respect to land–sea governance because of a lack of medium-term approaches, and lacks medium data collected and analysis systems. While integrated coastal management (IZCM), climate planning (CPP), and urban/regional marine special planning (MSP) are tools that can be useful, they are discussed only at the strategic level and are not sufficiently implemented at the grassroots level. It is an indisputable fact that the increasing human impact has left a tangible and visible residual imprint on the Earth's climate. For scientific researchers, 'sustainable development', 'climate change', and 'environmental protection' have gained increasing acceptance in recent years, while 'environmental management' is more frequently seen today [3]

The city of Constanta, located in the southeastern part of Romania, on the Black Sea's coasts, has as its main goal the growth of tourism, with mass tourism as an option. This is evident in both the paper Integrated Strategy for Sustainable Tourism Development in

Constanta County, 2019–2028 [4], and the Master Plan for the Protection and Rehabilitation of the Coastal Area [5]. The basis for medium- and long-term city development planning are laid out in these strategic development publications. There are, however, studies that examine the impacts of mass tourism on a town which draw attention to the traffic and pollution issues that this activity causes [6]. This paper is structured as follows: Section 2 describes the Constanta area regarding the tourism activity, Section 3 presents the methods used in this research, followed by results, and the last section presents the conclusion.

## 2. Case Study

For this study, we focus on the Constanta area of Romania. Constanta is an open sea city, located in the east of Romania, as shown in Figure 1. Constanta is one of the most important cities due to national/international tourism to the Black Sea. Due to its physico-geographical characteristics, the Romanian coastline has benefited greatly from the largest sums of tourist attention in the first half of the 1960s and in terms of attendance amenities in this time period. Due to this arrangement, Constanta was found to be the county with the most structures and the most vacation options. Mamaia is one of the administrative districts of Constanta and one of the best beach resorts, with a 7 km-long beach which faces to the east [7].

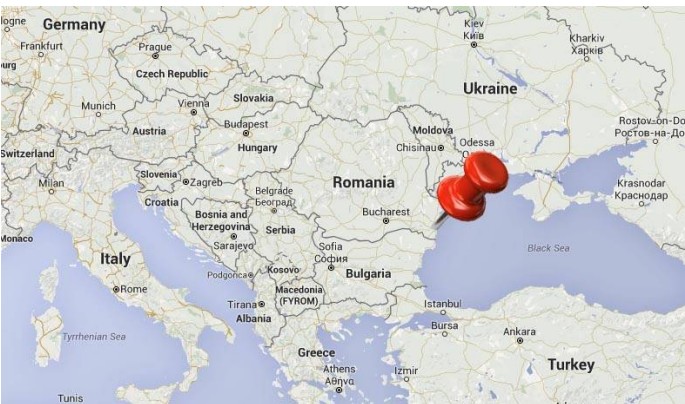

**Figure 1.** Location of Constanta City in Europe. Source: Google map processed image.

Even if tourism is a significant county-level activity, according to the town halls (Figure 2), the data indicate that the bulk of the operations are conducted on the shoreline of the Black Sea in Constanta County. Both the national tourism strategy [8] and the local strategic documents discussed in the preceding chapter specify a growth in tourism and accommodation capacity, as well as an improvement in the percentage of nights spent in accommodation units. This validates the concept of mass tourism as the only viable alternative for decision-makers. Simultaneously, one of the steps used by the authorities to enhance the number of tourists is to expand the surface of the beaches by constructing infrastructure. The formulation of infrastructure works in three phases is anticipated in the document Masterplan Protection and Rehabilitation of the Coastal Area, with the first phase already finished (in the period 2013–2015, with the surface of the beaches rising by over 120,000 square meters). Following that, we will look at the capacity of existing lodging in the area.

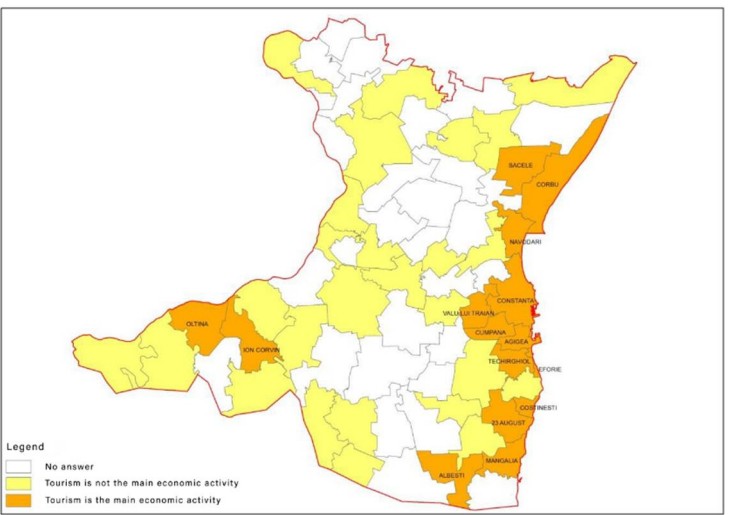

**Figure 2.** Localities in Constanta County where tourism is considered a main economic activity. Source: http://www.cjc.ro/dyn_doc/turism/Strategia_jud.Constanta_Faza2.pdf, accesed on 20 April 2021.

According to data from the National Statistical Institute, the total revenue generated by hotels, other lodging facilities, restaurants, and tourist agencies totaled RON 1,541,559,560 in 2017, accounting for 3.4 percent of the total revenue generated by all fields of activity in Constanta County. County-level tourism turnover increased by 56.7% between 2013 and 2017, but total business activity only slightly. If this type of business in 2013 held 2.2% of the market, it now has 3.4% of the market. (Table 1) [8–11].

**Table 1.** Turnover of active enterprises in Constanta County, from 2013 to 2017 (RON, 1 USD = 4.04 RON). Source: TEMPO-Online database (http://statistici.insse.ro:8077/tempo-online/#/pages/tables/insse-table) accessed on 10 June 2021.

|  | 2013 | 2014 | 2015 | 2016 | 2017 |
|---|---|---|---|---|---|
| **Total domains of activity, of which:** | 44,640,521,503 | 45,607,349,710 | 44,166,254,796 | 42,177,088,057 | 44,672,535,846 |
| **HR domain** | 983,700,816 | 1,086,868,232 | 1,399,752,043 | 1,460,529,931 | 1,541,559,560 |

Tourist structure.

- Tourist reception structures with accommodation functions.
- Tourism benefits from tourist accommodation. This is why the resort's location mostly controls the capacity of tourist flows.

Due to its physico-geographical characteristics, the Romanian coastline has benefited greatly from the largest sums of tourist attention in the first half of the 1960s and in terms of attendance amenities in this time period. Due to this arrangement, Constanta was found to be the county with the most structures and the most vacation options [12].

According to the latest official data for 2018, 834 units (9.9% of the national total) were open in Constanta County with a total of 84,891 places (24.0% of the national total) (Table 2).

**Table 2.** Evolution of accommodation capacity at the level of Constanta County and for the total country. Source: TEMPO-Online database (http://statistici.insse.ro:8077/tempo-online/#/pages/tables/insse-table) accessed on 10 June 2021.

| Years | Constanta County | | Romania | | % of Total Romania | |
|---|---|---|---|---|---|---|
| | Units | Places | Units | Places | Units | Places |
| **2012** | 738 | 84,690 | 5821 | 301,109 | 12.7% | 28.1% |
| **2013** | 745 | 85,756 | 6009 | 305,707 | 12.4% | 28.1% |
| **2014** | 746 | 87,496 | 6130 | 311,288 | 12.2% | 28.1% |
| **2015** | 755 | 87,848 | 6821 | 328,313 | 11.1% | 26.8% |
| **2016** | 761 | 85,285 | 6946 | 328,888 | 11.0% | 25.9% |
| **2017** | 838 | 84,157 | 7905 | 343,720 | 10.6% | 24.5% |
| **2018** | 834 | 84,891 | 8449 | 353,308 | 9.9% | 24.0% |

A report by the National Statistical Institute shows the following reality in Constanta County for the past seven years:

- The number of lodging structures increased significantly in 2018, with 13.0 percent more units than in 2012.
- The number of available accommodations has remained relatively constant.
- When compared to 2012, operational accommodation capacity (number of places–days) decreased by 2.0 percent in 2018.

The situation is explained by the fact that the new structures, while more numerous, are still much smaller in size (pensions, villas) in comparison to the large units that have emerged from the tourist circuit for various reasons (camping, camps, hotels) [13–15]. Additionally, the reduced capacity in all vacation properties also means reduced availability of accommodation, and in particular, the large hotels in tourist destinations are made available for less time in operation year to year. As was researched in Constanta County, the average duration of accommodation structures was approximately 125.8 days in 2012, and decreased to 124 days in 2017. By comparison, the national average is about 255 days [16].

Data retrieved from the government tourism agencies are considerably different than that found in the National Statistical Institute itself. According to the December 2018 data, there are 1807 tourist reception structures with accommodation functions in Constanta County, totaling 122,262 places. In comparison to the rest of the country, Constanta County has 13.4 percent of the total number of accommodation structures and 27.2 percent of the total number of places to stay, making the county of Constanta, and particularly the seaside area of the county, the most important tourist destination in the country (Figures 3 and 4).

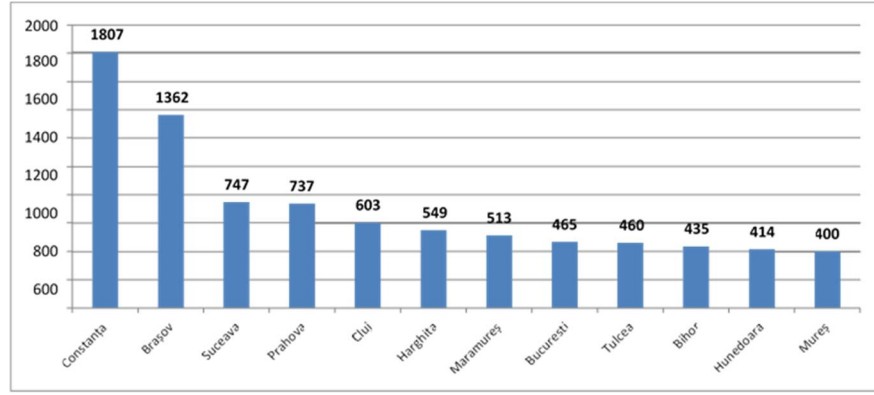

**Figure 3.** Number of tourist reception structures with accommodation functions, by county (December 2018). Source: http://turism.gov.ro/web/autorizare-turism/, accessed on 20 April 2021.

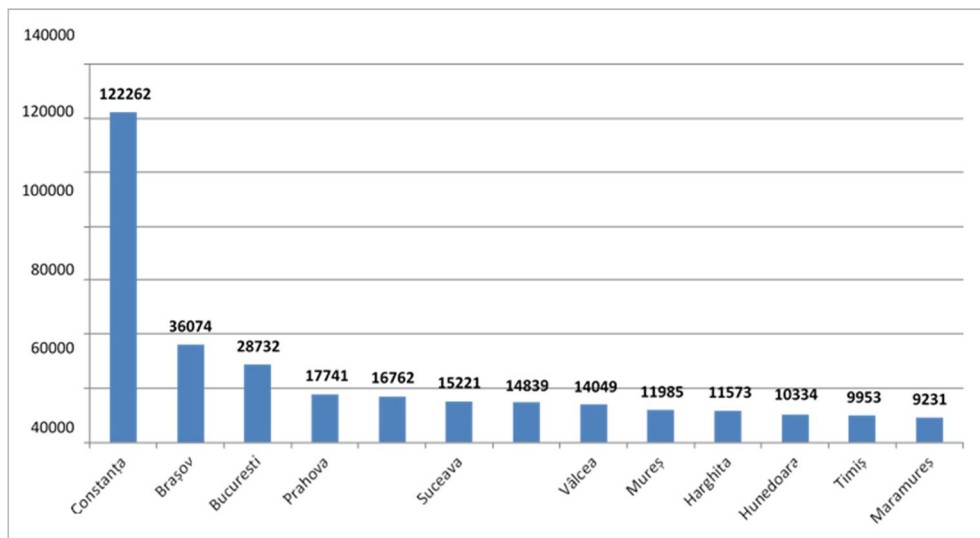

**Figure 4.** Number of accommodations, by county (December 2018). Source http://turism.gov.ro/web/autorizare-turism/, accessed on 20 April 2021.

In Table 3 is presented the accommodation capacity of Constanta County. The first main resorts on the Romanian Black Sea coast by number of places are: Mamaia (20.3%), Eforie Nord (14.6%), and Costineşti (13.0%).

**Table 3.** Accommodation capacity in Constanta County, by locality (December 2018). Source: http://turism.gov.ro/web/autorizare-turism/, accessed on 20 April 2021—processed data.

| Localities | Accommodation Structures | | Accommodation Places | |
|---|---|---|---|---|
| | Number | Percent | Number | Percent |
| **Constanța** | **309** | **17.1%** | **28.396** | **23.2%** |
| Constanța | 87 | 4.8% | 3.519 | 2.9% |
| Mamaia | 222 | 12.3% | 24.877 | 20.3% |
| **Mangalia** | **258** | **14.3%** | **32.904** | **26.9%** |
| Mangalia | 29 | 1.6% | 1.683 | 1.4% |
| Cap Aurora | 7 | 0.4% | 2.184 | 1.8% |
| Jupiter | 36 | 2.0% | 5.848 | 4.8% |
| Neptun-Olimp | 88 | 4.9% | 9441 | 7.7% |
| Saturn | 40 | 2.2% | 6.438 | 5.3% |
| Venus | 58 | 3.2% | 7.310 | 6.0% |
| **Eforie** | **431** | **23.9%** | **23.857** | **19.5%** |
| Eforie Nord | 318 | 17.6% | 17.795 | 14.6% |
| Eforie Sud | 113 | 6.3% | 6.062 | 5.0% |
| **Costinești** | **355** | **19.6%** | **15.850** | **13.0%** |
| **Limanu** | **127** | **7.0%** | **4.177** | **3.4%** |
| 2 Mai | 38 | 2.1% | 1.047 | 0.9% |
| Vama Veche | 88 | 4.9% | 3.084 | 2.5% |
| Limanu | 1 | 0.1% | 46 | 0.0% |
| **Năvodari** | **226** | **12.5%** | **13.114** | **10.7%** |
| **Techirghiol** | **28** | **1.5%** | **1.390** | **1.1%** |

**Table 3.** *Cont.*

| Localities | Accommodation Structures | | Accommodation Places | |
|---|---|---|---|---|
| | Number | Percent | Number | Percent |
| **Other localities** | **73** | **4.0%** | **2.574** | **2.1%** |
| Adamclisi | 1 | 0.1% | 30 | 0.0% |
| Agigea | 10 | 0.6% | 204 | 0.2% |
| Aliman | 2 | 0.1% | 28 | 0.0% |
| Băneasa | 1 | 0.1% | 20 | 0.0% |
| Cernavodă | 4 | 0.2% | 142 | 0.1% |
| Cobadin | 1 | 0.1% | 10 | 0.0% |
| Crucea | 1 | 0.1% | 54 | 0.0% |
| Cumpăna | 1 | 0.1% | 18 | 0.0% |
| Lipnița | 2 | 0.1% | 42 | 0.0% |
| Lumina | 1 | 0.1% | 59 | 0.0% |
| Medgidia | 6 | 0.3% | 188 | 0.2% |
| Mihail Kogălniceanu | 1 | 0.1% | 36 | 0.0% |
| Oltina | 1 | 0.1% | 16 | 0.0% |
| Ostrov | 3 | 0.2% | 74 | 0.1% |
| Ovidiu | 3 | 0.2% | 63 | 0.1% |
| Saligny | 1 | 0.1% | 12 | 0.0% |
| Seimeni | 1 | 0.1% | 32 | 0.0% |
| 23 August | 20 | 1.1% | 1.314 | 1.1% |
| Corbu | 13 | 0.7% | 232 | 0.2% |
| TOTAL | **1.807** | **100%** | **122.262** | **100%** |

## 3. Materials and Methods

The multi-criteria model was applied to mass tourism in Constanta using the PESTEL technique. This model accounted for the impact of relevant indices on all three major factors: economic, social, and environmental. These processes can be applied to any residential or business destination, especially in a touristic or coastal area, identifying the activities and considering each location's unique characteristics [17–20].

It is important to run multicriteria analyses over time to identify possible changes in the factors. The more in-detailed the analysis you give, the more time you have to identify and reduce security risks that overlap with the various variables.

Indicators of activity in the area, as well as their impact on other economic, social, and environmental activities, were identified. The different perspectives were evaluated on the basis of the PESTEL model [21–25]; for Constanta, used three parameters of analysis:

Economic factor: In terms of agriculture and aquaculture, the analysis was carried out while taking into account local products and their market access. Despite being a predominantly agrarian country, Romania's trade balance still favors food imports, according to existing data at the Institute of Statistics. The occupancy of hotels and restaurants, as well as the economic use of the coastal beach, were the two indicators considered in this study. We also considered the activity of retailers in the studied area. In this chapter, we used data from the retail industry's turnover and profit from 2011 to 2019. Car traffic, use of local transportation, and electricity consumption are also important indicators of the stress that the community faces during the summer season. For real-time traffic monitoring, we used Google Maps' traffic function, and for relevant traffic data on the number of vehicles in a given area, we used TOMTOM's free data.

Social factor: This took into account the social environment and identified the trends that existed in the analyzed area. To examine this factor, two indicators were developed: employment and crime.

Environmental factor: This factor referred to the impact of other factors on the environment as well as the impact of ecological aspects. The environmental factor includes

indicators for air, water (sea water and fresh water), waste (food waste and general waste), and protected areas.

Consideration of only the proposed indicators cannot lead to sustainable tourism activity. It is important to identify which of the variables add more or less pressure on some of the other variables. In fact, the supportability factor was applied, taking into account how the supportability of these activities would compare to each other. The tipping point occurs where one activity must stop before the other disappears. For activities that harmed one another, their credibility rating was reduced to zero if the percentage of total activity was small. Similarly, supportability has an effect on every other activity, and is inversely proportional to it. A rough estimate of the extent of the damage was based on existing data from government sources [26].

In the multi-criteria analysis, we used two scenarios: scenario number one, which refers to a moderate number of tourists, and scenario number two, which depicts the actual number of tourists. The relationship between the indicators and the number of tourists was established by using coefficients for the three months of the season under consideration, namely June, July, and August.

The coefficients added for the summer months were between 1.5 and 2.5, based on the number of tourists present and the population of the constant city.

In multi-criteria analysis, some of the indicators remained constant. This was due to the fact that mass tourism had little impact on them. In these cases, the indicators were labeled with the number 1, indicating that they were at the positive end of the analysis. For example, if we consider the consumption of local food products, because Romania is already an importer of food products, it follows that local producers with a limited stock of products exhaust their stocks regardless of tourist traffic. In general, retailers rely on suppliers who bring in imported goods to meet their food needs during the summer season. Furthermore, during the summer season, the workforce is typically relocated from one unit located in the city to another unit located on the coast, both units belonging to the same economic agent. The Statistical Institute's employment figures clearly show that the unemployment rate does not differ significantly between the seasonal and off-season months [27,28]. This demonstrates, on the one hand, the relocation of the workforce between units, and, on the other, the hiring of personnel from outside Constanta, with the newcomers having a negligible economic impact on the local community.

Returning to the method of calculating the indicators, this took the form of:

$$i_{mt} = i_{wt} \times c_{nt}/100 \tag{1}$$

where:

$i_{mt}$ = *indicator that supports mass tourism.*
$i_{wt}$ = *indicator that does not support mass tourism.*
$c_{nt}$ = *coefficient for the number of tourists.*

Another equation was used to calculate the indicator for the retailers' indicator. This was:

$$i_{mt} = (Pn/Rn) \times c_{nt} \tag{2}$$

where:

$Rn$ = *net result.*
$Pn$ = *net profit.*
$c_{nt}$ = *coefficient for the number of tourists.*

A linear regression was used to predict the values until 2050 for different factors of the analyzed model. The values were predicted based on the second variable in the simple linear regression. The predicted values are presented as a function of the other variable in the form of a straight line in the simple linear regression of this section. Linear regression is used to find the right line through the points that is best suited to the data. The best line is also known as a regression line.

## 4. Results

Based on those presented in the previous section, we analyzed the evolution of all three factors—economic, social, and environment—based on both scenarios—with mass tourism and without mass tourism—for summer time—months June, July, August—in the period 2011–2015. To view the future effects on these factors, we extended the analysis until the year 2050.

Figure 5 depicts the evolution of passenger transportation in the city of Constanta under scenarios 1 (no mass tourism) and 2 (with mass tourism). There is a noticeable difference in the efficiency of public passenger transportation during the months when tourists visit the coast. At this point, we can say that local policies should focus on encouraging residents to use public transportation more frequently, so that public transportation activity is more efficient during the off season. This indicator represents a situation in which mass tourism is a positive factor in the local community's development.

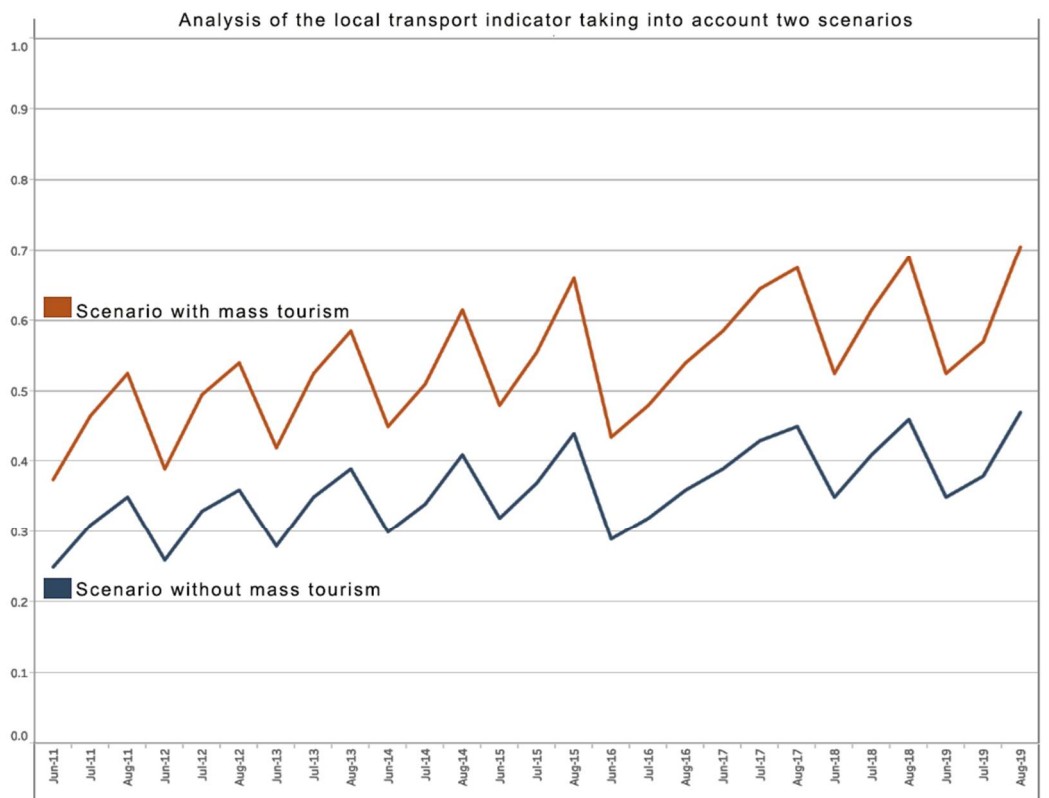

**Figure 5.** Local transport activity in the two scenarios.

By projecting local transportation to 2050 (Figure 6), it is possible to see that, based on the current data introduced into the system, the mass tourism hypothesis is more beneficial, reaching an optimal around 2039, whereas the assumption without mass tourism reaches an optimal of 0.84 out of 1 at the level of 2050. The trend is clear in both scenarios, and the optimization of this activity is at the disposal of the authorities, who can take measures to stimulate this indicator.

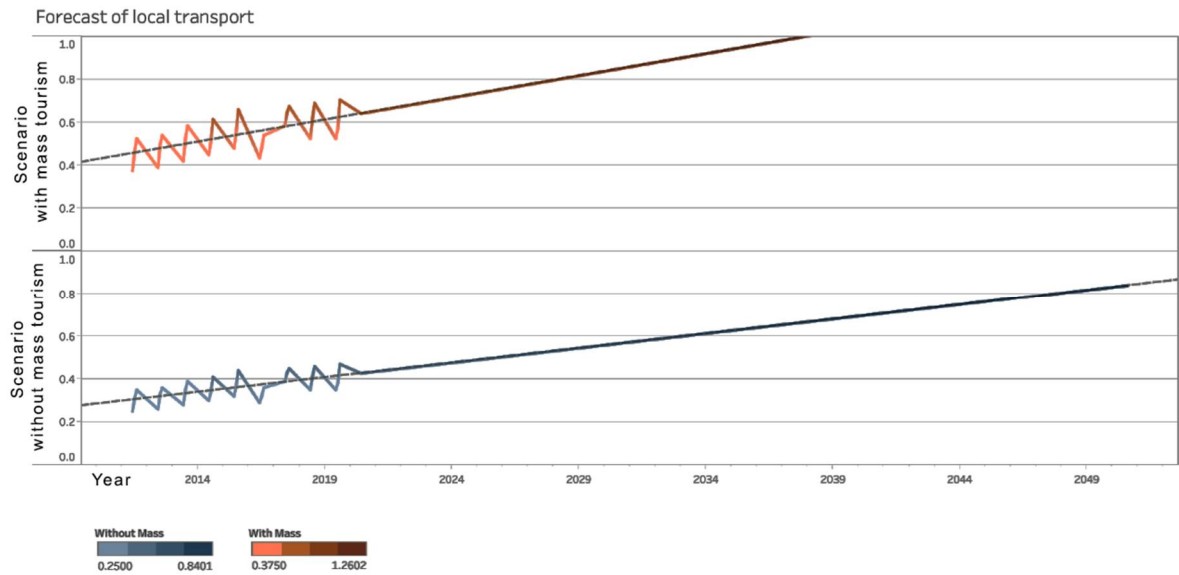

**Figure 6.** Forecast of local transport activity by 2050.

Similarly, we have expanded the analysis of economic factors (Figures 7 and 8), social factors (Figure 9), and environmental factors (Figure 10) to account for the two scenarios, one with and one without mass tourism.

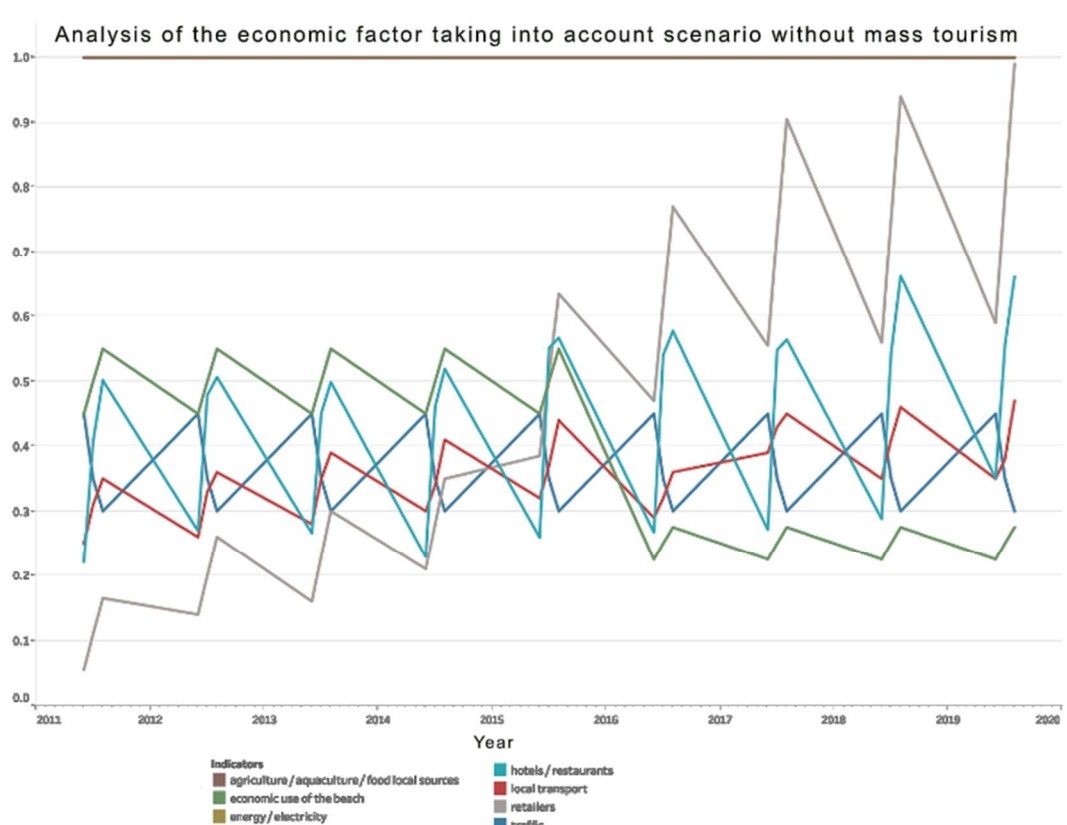

**Figure 7.** Analysis of economic factor indicators for the without mass tourism scenario.

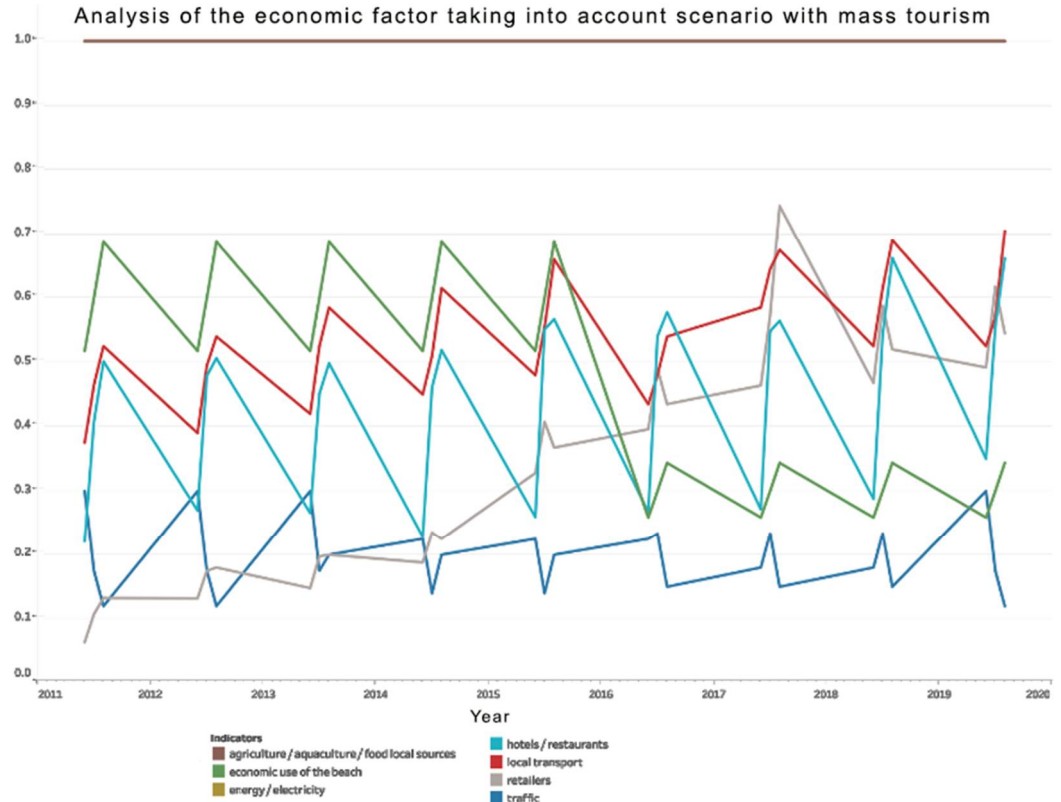

**Figure 8.** Analysis of economic factor indicators for the mass tourism scenario.

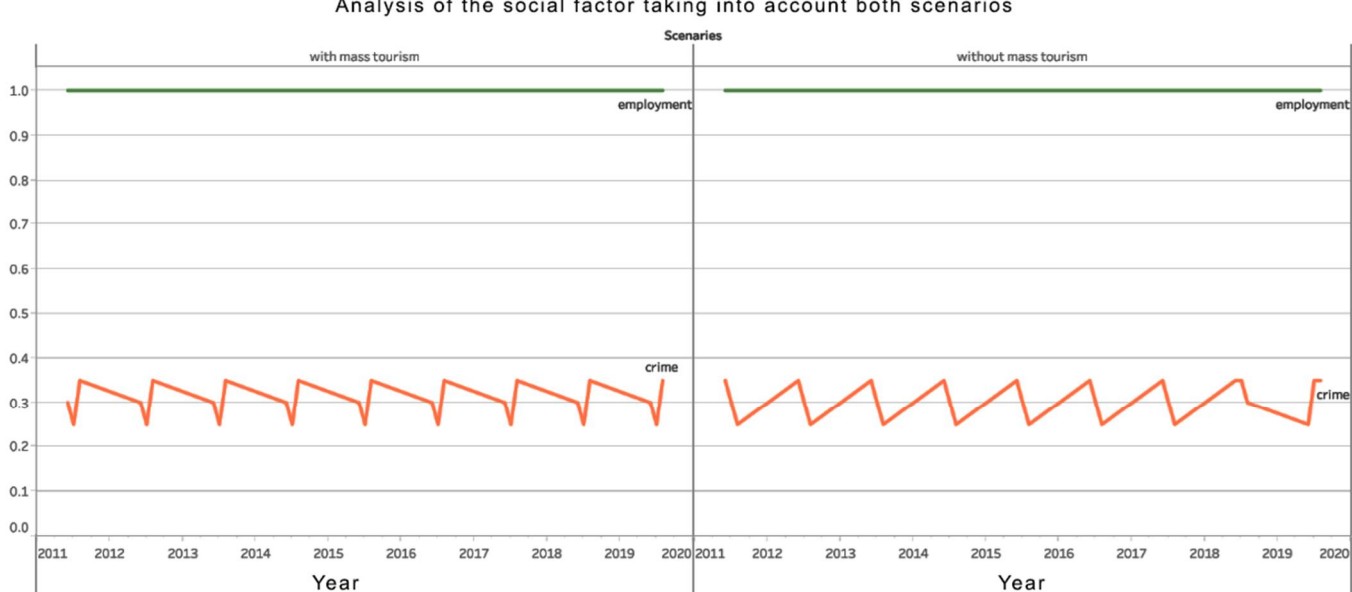

**Figure 9.** Analysis of social factor indicators for the two scenarios.

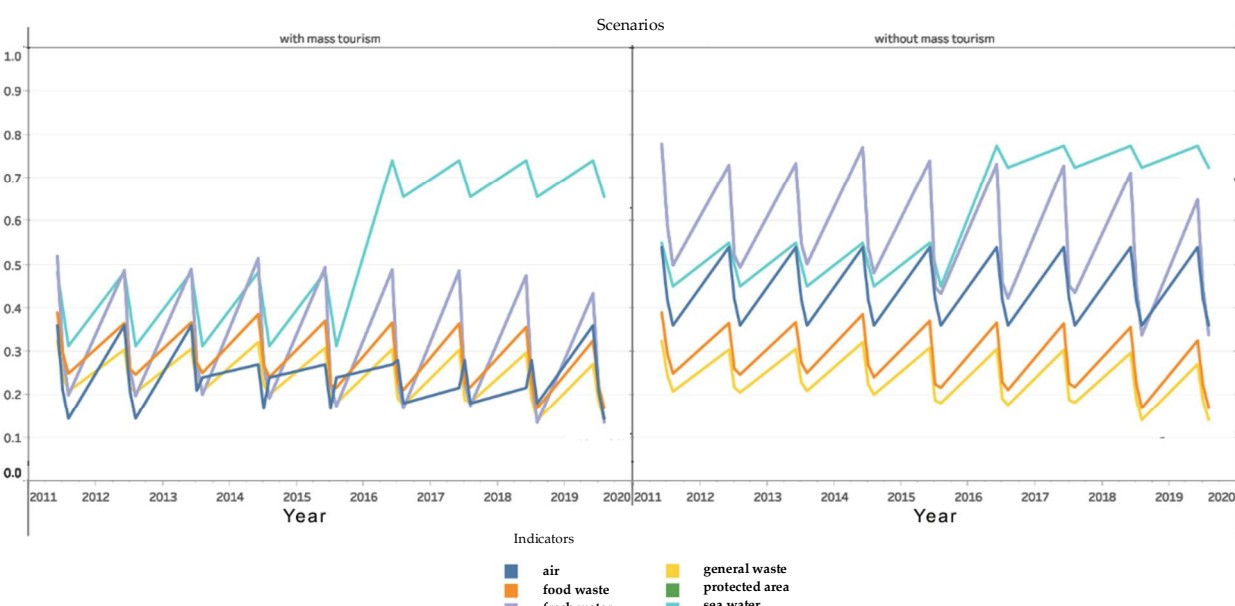

**Figure 10.** Analysis of environmental factor indicators for the two scenarios.

The evolution of indicators oscillates, with peaks occurring most frequently in August of each year, with this month being the most dynamic. In the case of the economic factor under consideration, several relevant indicators were considered, including the consumption of local products from agricultural or aquaculture sources, the activity of retailers in public food, the activity of the hotel and restaurant sector, local transportation and traffic, electricity consumption, and the economic use of the beach. This last indicator was added because, according to existing studies and strategies, the beach is the primary destination of seaside tourism, and the Romanian government has invested significantly in this goal. The best economic use of the beach was calculated based on the number of tourists and locals who use the beach's surface. The optimal area required for a beach tourist, which is 4 square meters, is established by both domestic legislation and international standards. In these circumstances, the optimal economic use of the beach for us was that the beach must be suitable in terms of area for the number of existing visitors, with a beach with an insufficient or excessively large area having a negative impact on tourist activity.

In the case of the social factor, we only looked at two indicators: labor force participation and criminality during the summer season. As previously stated, we discovered that the workforce used is especially the one relocated from the city between the units of the same operator, or the workforce that comes from other localities, by analyzing the data provided by the Statistical Institute, but also based on other studies and interviews with factors involved. In terms of criminality, it is high during the summer season but fairly consistent from year to year.

We considered the air quality, which was primarily related to the increased traffic on the coast during the summer season, the amount of waste produced (food and general), the quality of bathing water, the quantity and quality of drinking water, and the impact of tourists on protected areas. In terms of waste production, the data were incorporated into the county waste management plan, which shows a significant increase (on average 30 percent) in the amount of waste produced during the summer season, waste produced by tourists. The Statistics Institute also publishes an annual report on food waste, which shows how much food each novel wastes each year. In terms of bathing water quality, the Public Health Directorate's annual reports to the European Commission show that the bathing water is in the optimal parameters, with moderate, good, and very good status. The reduced use of the beach also helped to improve the quality of the bathing water.

However, it is clear that, while the statistics are positive, more detailed analyses on this parameter are required, given the numerous cases of pollution and discomfort reported in recent years. However, these are not the topic of this work. In terms of protected areas, the most significant protected area within a reasonable distance of Constanta is Lake Siutghiol, which is an avifaunistic protected area with significant ecological activity, particularly for migratory birds during the winter. Because the influence of tourist activity during the summer season is negligible, the indicator does not have extremely low values. We did not examine the degree of urbanization in the protected area in this paper because we believe that tourist activity during the summer does not have a significant impact on migratory bird wintering activity.

By achieving an average of the economic factor indicators (Figure 11) between 2011 and 2019, it can be seen that, while mass tourism was initially more beneficial, the situation changed in the middle of the period, and the activity without mass tourism became the better option for the community. A projection of this trend until 2050 (Figure 12) shows that both options are economically beneficial to the community, because the trend is growing; however, the growth in the option without mass tourism is more pronounced, reaching an optimal around 2048, whereas the optimal in the other option is expected after 2055.

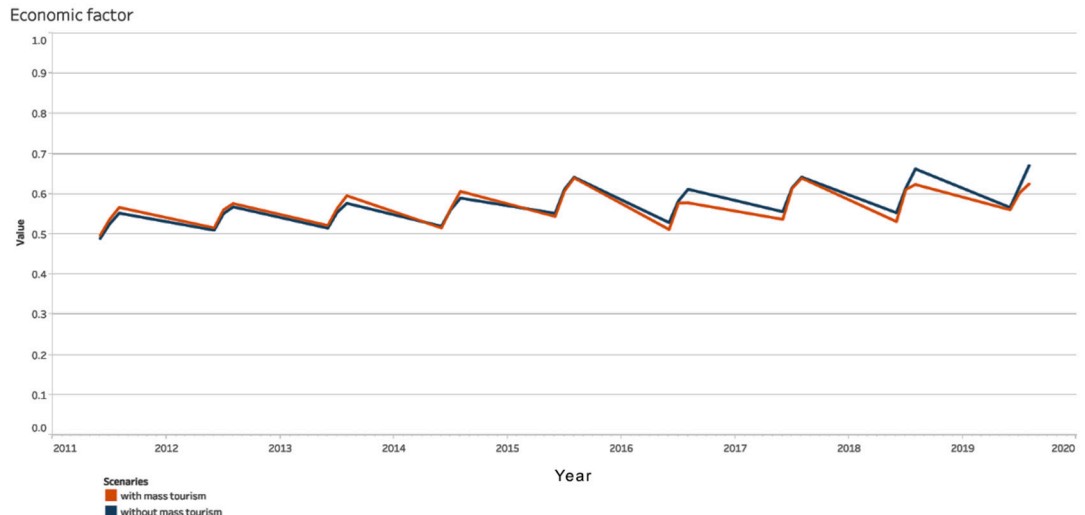

**Figure 11.** Analysis of economic factor average indicators for the two scenarios.

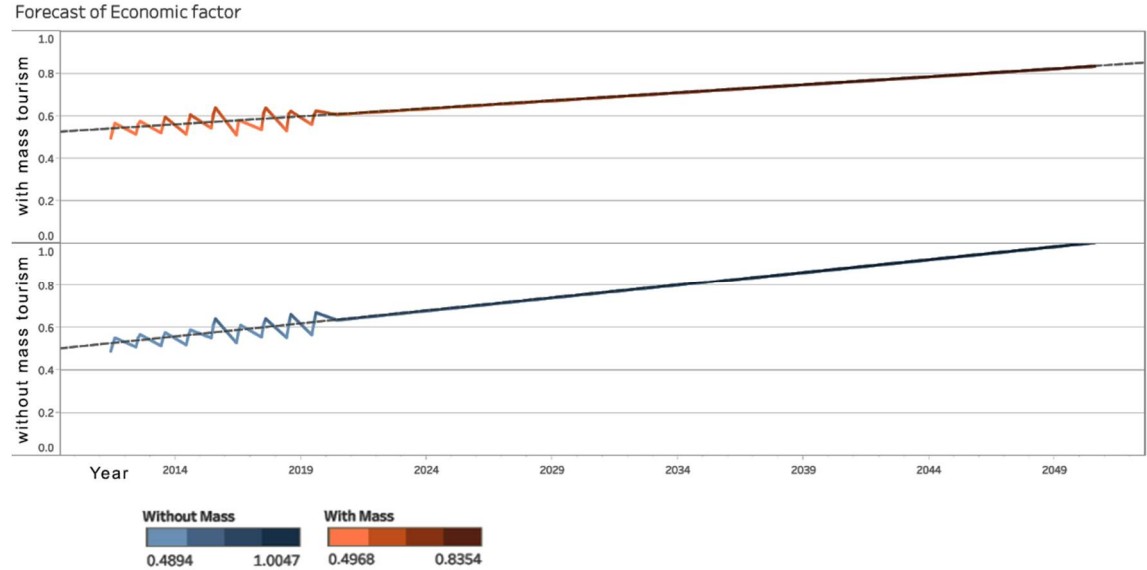

**Figure 12.** Economic factor forecast by 2050.

We conducted a similar analysis in terms of the social factor. If we focus on the intercalation of the two scenarios in the studied period, 2011–2019 (Figure 13), the situation becomes more interesting in terms of the forecasts of the two scenarios. The best option is also proven in the situation without mass tourism, with a higher value in 2050 than in the scenario with mass tourism. On the other hand, the trend is decreasing in terms of the option without mass tourism, while the trend is increasing in terms of the option with mass tourism (Figure 14). It is expected that in terms of community benefits, a choice will be made between the two options between 2050 and 2060. This social factor analysis provides some arguments for planners and public policymakers to choose one scenario over another based on the community's development directions. Simultaneously, the analysis can be improved by including additional social indicators such as education, access to culture, health, and so on.

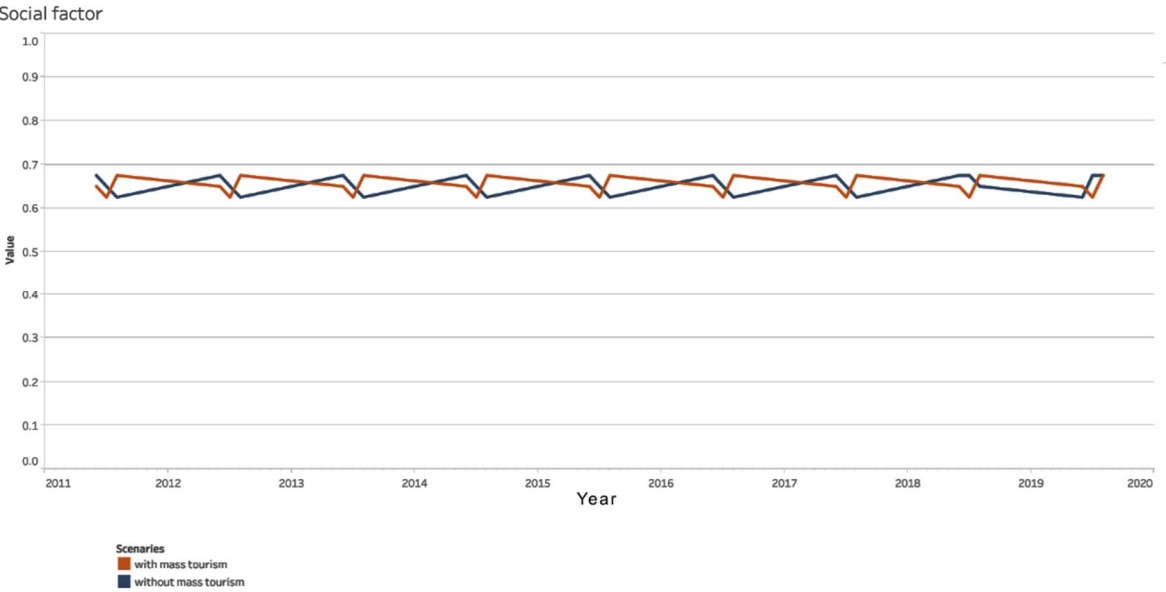

**Figure 13.** Analysis of social factor average indicators for the two scenarios.

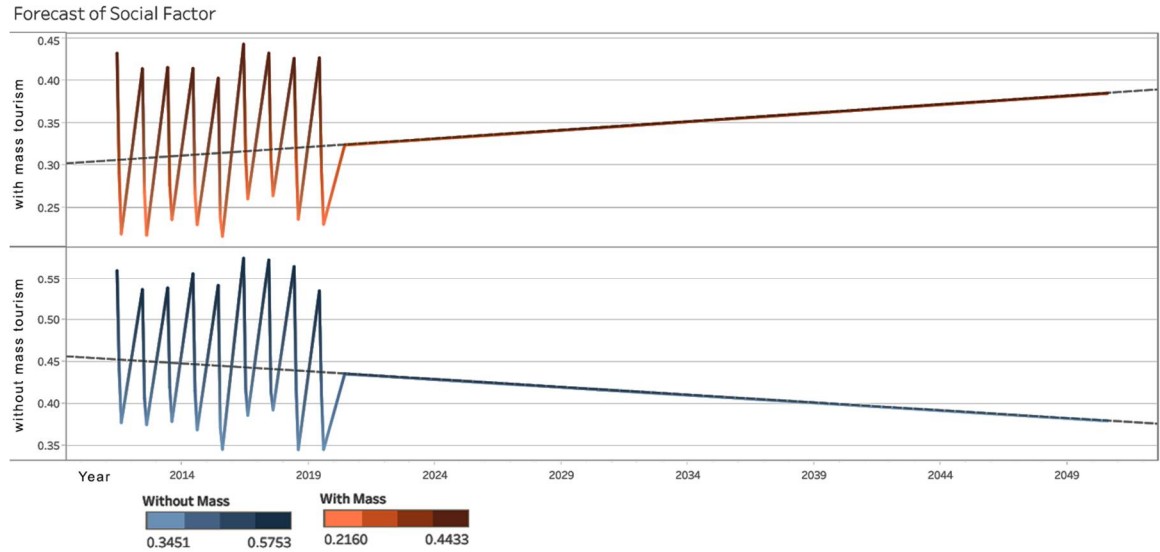

**Figure 14.** Forecast of the social factor until 2050.

In terms of the environment, an analysis of the averages of the proposed indicators for the period 2011–2019 (Figure 15) reveals that the scenario without mass tourism has values

that are higher than the scenario with mass tourism. In terms of trends, the forecast until 2050 is particularly interesting, with the trend without mass tourism slightly decreasing and the trend with mass tourism slightly increasing (Figure 16). However, the main option remains one without mass tourism, and if current conditions continue, the option without mass tourism will be the best scenario for the community even after 2050.

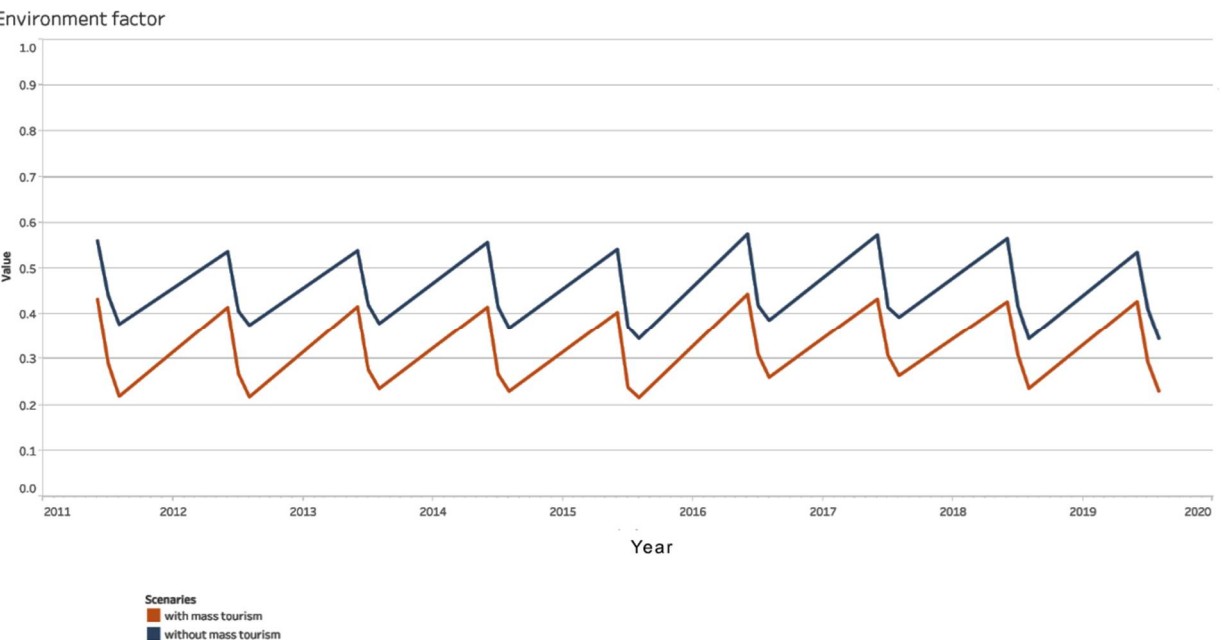

**Figure 15.** Analysis of environmental factor average indicators for the two scenarios.

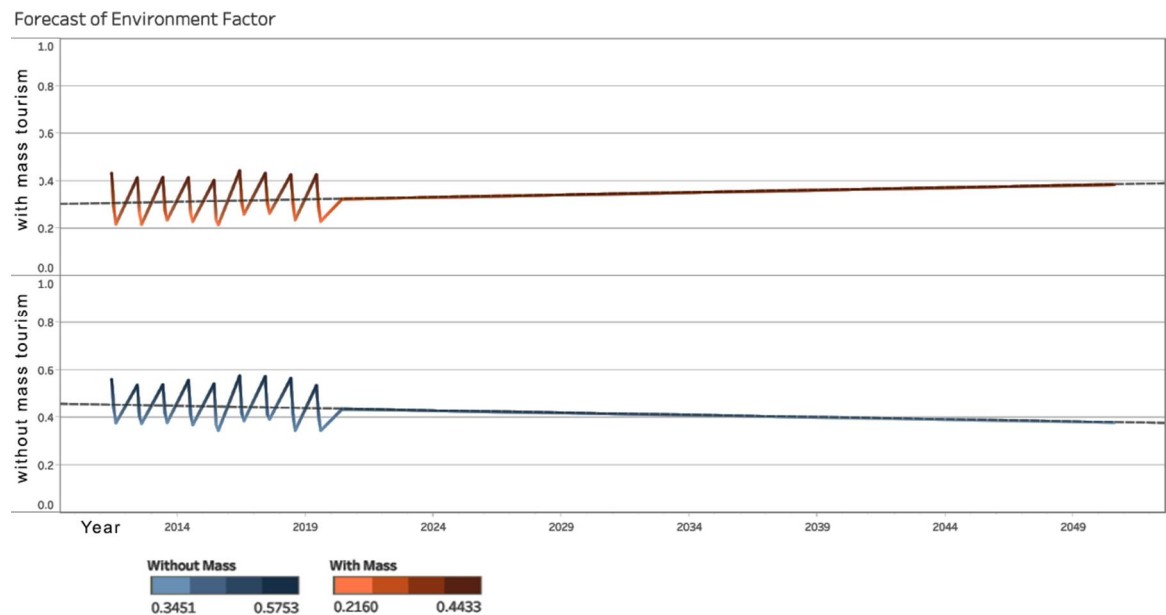

**Figure 16.** Average factor forecast by 2050.

The final part of the analysis was completed by calculating the average of the two scenarios for all of the proposed factors (economic, social, and environmental). As shown in Figure 17, the scenario without mass tourism outperforms the scenario with mass tourism over the period 2011–2019. This demonstrates that the previous option was not the best fit for the community. Regarding the projections of the two scenarios until 2050, the trends in both situations are increasing (Figure 18). The scenario without mass tourism has faster growth, but the difference between the two scenarios is not significant. This situation

presents a challenge for policymakers seeking to improve local policies in accordance with the path they wish to take for community development.

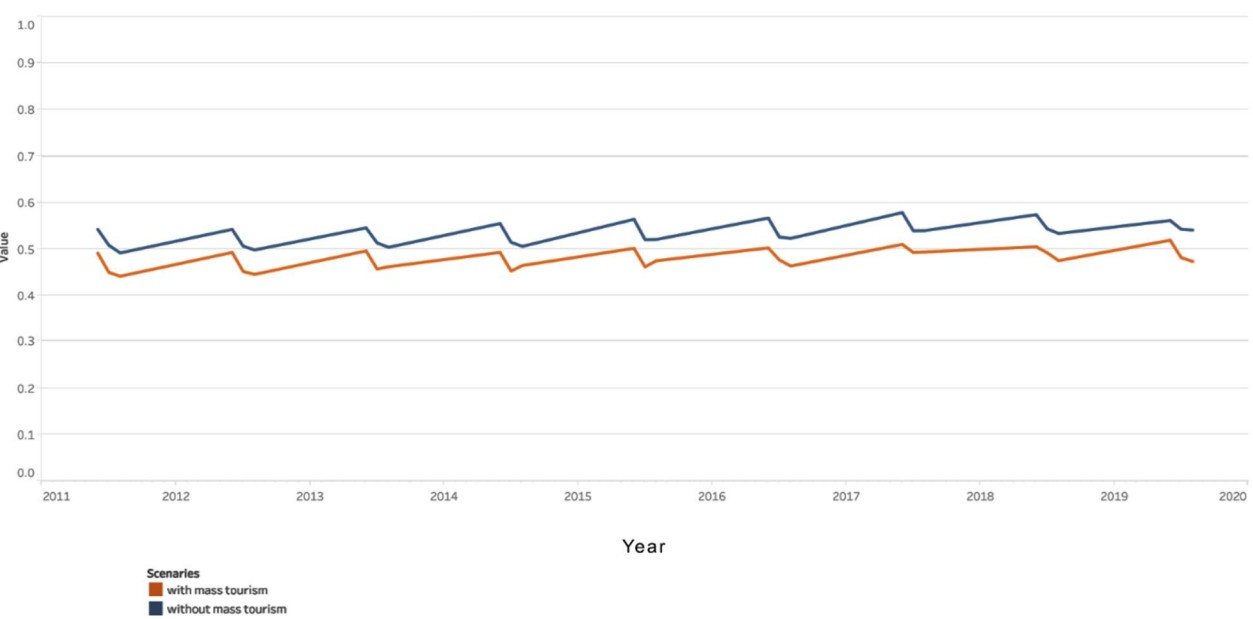

**Figure 17.** Analysis of indicator averages for the two scenarios.

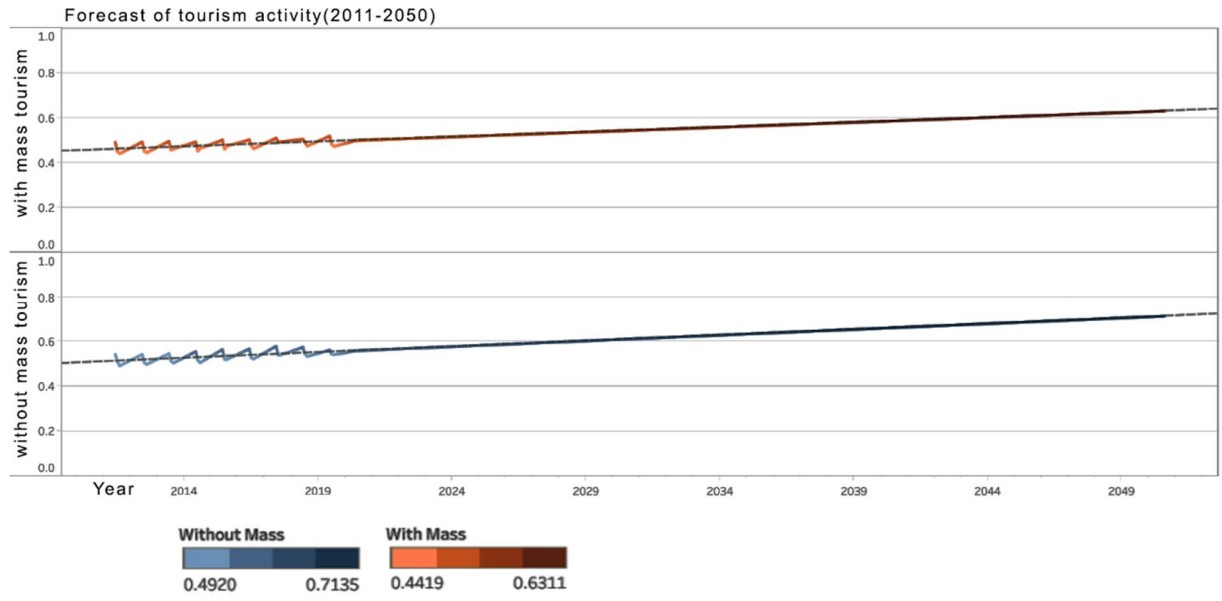

**Figure 18.** Forecast of indicator averages for the two scenarios for 2011–2050.

## 5. Conclusions

The multi-criteria analysis of the mass tourism management model in terms of its impact on the local community in Constanta is a complex tool that can aid in the development of local policies. Depending on the pressures exerted by various activities on other activities taking place in the community, the scenarios considered may or may not be viable. The accuracy and consistency of the data used in this analysis are critical. Furthermore, the correct identification of indicators and their interrelationships is an important process in carrying out a correct analysis. Last but not least, determining the calculation coefficient for the activity under consideration is critical, because bringing together data from various activities can be difficult when attempting to identify a viable model.

The PESTEL methodology was used to apply the multicriteria model to mass tourism in Constanța. The model's use demonstrated the impact of relevant actions on three primary factors: economic, social, and environmental. By recognizing existing activities and determining the relationships between these activities, such a model can be applied to any tourist or seaside location.

Multicriteria analysis must be performed throughout time in order to detect probable changes in components and provide more accurate model predictions.

Only considering the recommended indicators will not result in a long-term tourism activity. It is crucial to figure out which variables exert more or less pressure on the others. The affordability index must be determined for this. The equilibrium point must be determined, where one activity's pressure must cease before the other activity vanishes.

In the case studied in this paper, we assumed that the number of tourists visiting a location has a significant impact on the local community. Although it appears that mass tourism is beneficial from an economic standpoint, it can be observed that this type of tourism has a negative impact on some economic and environmental indicators, and the positive impact is often insignificant. This is not to say that a community cannot aim to use mass tourism as a form of tourism. It is just that in these cases, they must identify the high-pressure activities and take measures to relax the relationships between them.

One of the study's limitations was the data's availability and accuracy. Although most data are collected by public entities, they are not made available to the broader public. Furthermore, data gathering methods differ, and their accuracy in many circumstances is low. Another noteworthy limitation was the number of tourists visiting the area during the summer, even though the area is well known for the fact that many local people offer their own houses for accommodation. There are no relevant statistics on the number of tourists staying in these facilities. Another limitation of this study is that the activities in the coastal area are far more numerous and complex. As a result, in order to examine an effective situation, the interaction between these activities must be discovered, as well as the tolerance threshold between them. Last but not least, the study is time-limited in the sense that this analysis must be refreshed after a given period to determine whether the data entered are still relevant.

**Author Contributions:** Conceptualization, C.A. and A.-E.M.; methodology, A.-E.M.; validation, C.A., A.-E.M. and E.R.; formal analysis, C.A.; investigation, A.-E.M.; resources, A.-E.M. and E.R.; data curation, C.A.; writing—original draft preparation, C.A.; review and editing, E.R.; visualization, C.A.; supervision, A.-E.M. and E.R.; project administration, A.-E.M.; funding acquisition, E.R. All authors have read and agreed to the published version of the manuscript.

**Funding:** This work was carried out in the framework of the research project DREAM (Dynamics of the REsources and technological Advance in harvesting Marine renewable energy), supported by the Romanian Executive Agency for Higher Education, Research, Development, and Innovation Funding—UEFISCDI, grant number PN-III-P4-ID-PCE-2020-0008.

**Institutional Review Board Statement:** Not the case this study does not involve humans or animals.

**Informed Consent Statement:** Not the case this study does not involve humans or animals.

**Data Availability Statement:** The data that support the findings of this study are available in the public domain.

**Acknowledgments:** The results of this work will be also presented to the 9th edition of the Scientific Conference organized by the Doctoral Schools of "Dunărea de Jos" University of Galati (SCDS-UDJG) http://www.cssd-udjg.ugal.ro/ (accessed date 20 April 2021) that will be held on the 10 and 11 June 2021, in Galati, Romania.

**Conflicts of Interest:** The authors declare no conflict of interest.

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
