# Peer review of "Multi-Criteria Analysis of the Mass Tourism Management Model Related to the Impact on the Local Community in Constanta City (Romania)"

_inventions, doi:10.3390/inventions6030046_

Round 1

Reviewer 1 Report

I think this study is not supported theoretically. Missing a theoretical review chapter to support how tourism is important to the local economy. tudies can be put as examples. No relevant references

However, I found the case study very good and interesting

I think the conclusions are very brief. There should be some theoretical conclusions. This chapter "conclusions" needs to be expanded.

Indicate the limitations of this research. Also if there are future lines of research.

I encourage the authors to make the appropriate changes and revisions so that the manuscript is well structured, since the practical case seems interesting to me.

Author Response

ANSWERS TO THE REVIEWERS' COMMENTS 

Manuscript ID: Inventions- 1245547

GENERAL COMMENTS

A revision has been carried out following carefully all the indications, suggestions and observations formulated by the reviewers.

The main changes operated are outlined next:

1) Detailed explanations have been provided at various points in order to make the results of the work more comprehensible.

2) Some new references have been added.

At this point, it has to be also highlighted that the authors tried to follow all the suggestions and observations formulated by the reviewers and to operate (as much as it was possible) all the corrections indicated by them. Furthermore, in order to follow the corrections operated in the paper, a version of the manuscript having all the changes operated tracked (with the option track changes) has been also uploaded together with the last form of the manuscript (without the changes tracked).

The specific corrections operated according to the suggestions of the reviewers are given next together with detailed explanations.

 Reviewer # 1:

I think this study is not supported theoretically. Missing a theoretical review chapter to support how tourism is important to the local economy. Studies can be put as examples. No relevant references

Regarding this point, I would like to mention that the authors have consulted several strategic documents and studies in the field on the development of tourist activity in Constanta County as well as the impact of mass tourism on coastal communities. In this respect, we have made the following additions:

  • In the Introduction chapter we completed with the following paragraph:

The city of Constanta, located in the southeastern part of Romania, on the Black Sea's coasts, has as its main goal the growth of tourism, with mass tourism as an option. This is evident in both the paper Integrated Strategy for Sustainable Tourism Development in Constanta County, 2019-2028 [4] and the Master Plan for the Protection and Rehabilitation of the Coastal Area [5]. The basis for medium and long-term city development planning are laid out in these strategic development publications. There are, however, studies that examine the impacts of mass tourism on a town and draw attention to the traffic and pollution issues that this activity causes [6].

In the chapter Case study we filled with the following paragraph

Both the national tourism strategy [8] and the local strategic documents discussed in the preceding chapter specify a growth in tourism and accommodation capacity, as well as an improvement in the percentage of nights spent in accommodation units. This validates the concept of mass tourism as the only viable alternative for decision-makers. Simultaneously, one of the steps used by the authorities to enhance the number of tourists is to expand the surface of the beaches by constructing infrastructure. The formulation of infrastructure works in three phases is anticipated in the document Masterplan Protection and Rehabilitation of the Coastal Area, with the first phase already finished (in the period 2013-2015, with the surface of the beaches rising by over 120,000 square meters).

At the same time, we have added the following references:

  • Strategia integrata de dezvoltare durabila a turismului in judetul Constanta, in perioada 2019-2028
  • Masterplanul Protectia si reabilitarea zonei costiere
  • Ayhan Akis The effects of mass tourism: A case study from Manavgat (Antalya – Turkey)
  • Strategia naÅ£ionalăa României pentru dezvoltarea turismului. 2019-2030

However, I found the case study very good and interesting

  • Thank you for your considerations about our study

I think the conclusions are very brief. There should be some theoretical conclusions. This chapter "conclusions" needs to be expanded.

Indicate the limitations of this research. Also if there are future lines of research.

  • In the Conclusions chapter, we have completed the following paragraphs:
  • The PESTEL methodology was used to apply the multicriteria model to mass tourism in ConstanÈ›a. The model's use demonstrated the impact of relevant actions on three primary factors: economic, social, and environmental. By recognizing existing activities and determining the relationships between these activities, such a model can be applied to any tourist or seaside location.

Multicriteria analysis must be performed throughout time in order to detect probable changes in components and provide more accurate model predictions.Only considering the recommended indicators will not result in a long-term tourism activity. It's crucial to figure out which variables exert more or less pressure on the others. The supportability index must be determined for this. The equilibrium point must be determined where one activity's pressure must cease before the other activity vanishes.

  • One of the study's limitations was the data's availability and accuracy. Although most data is collected by public entities, it is not made available to the broader public. Furthermore, data gathering methods differ, and their accuracy in many circumstances is low. Another noteworthy limitation was the number of tourists visiting the area during the summer, despite the fact that the area is well-known for the fact that many local people offer their own houses for accommodation, but there are no relevant statistics on the number of tourists staying in these facilities. Another limitation of this study is that the activities in the coastal area are far more numerous and complex. As a result, in order to examine an effective situation, the interaction between these activities must be discovered, as well as the tolerance threshold between them. Last but not least, the study is time-limited in the sense that this analysis must be refreshed after a given period to determine whether the data entered is still relevant.

I encourage the authors to make the appropriate changes and revisions so that the manuscript is well structured, since the practical case seems interesting to me.

  • Thank you again for your feedback. We hope that the changes made meet the requirements expressed in the observations

Reviewer 2 Report

I have found the manuscript interesting. I think that the title is appropriate. The authors focus on study the mass tourist activity in Constanta through several conditions: social, economic, and environmental. The authors also make an effort to build a model and create a projection of the activities by 2050. These are the aims of the paper, which are clearly outlined both in the abstract and in the introduction section.

The abstract fulfills its function of summarizing the paper’s content, and it remarks on the aims of the study. I would suggest adding the key findings. 
I would also propose to improve the readability of the text, using the same terms (p.e. PESTEL or PESTLE), and defining the acronyms, abbreviations, or initialisms the first time they appear in the text (p.e. INS: I suppose it refers to National Statistical Institute).

There is repeated text in lines 99 to 103. 

Regarding table 1, I suggest expressing it in thousands or millions,  indicating the equivalence in euros or dollars. 

The conclusions section could be extended a bit more, and I miss a discussion section. In addition, the limitations of the study should be indicated, such as the fact that it is making thirty-year predictions with data from only five periods.

In general, the work seems interesting to me. The authors make a real contribution to the scientific literature with the analysis of the mass tourist activity.

I wish the authors good luck in revising their paper.

Author Response

ANSWERS TO THE REVIEWERS' COMMENTS 

Manuscript ID: Inventions- 1245547

GENERAL COMMENTS

A revision has been carried out following carefully all the indications, suggestions and observations formulated by the reviewers.

The main changes operated are outlined next:

1) Detailed explanations have been provided at various points in order to make the results of the work more comprehensible.

2) Some new references have been added.

At this point, it has to be also highlighted that the authors tried to follow all the suggestions and observations formulated by the reviewers and to operate (as much as it was possible) all the corrections indicated by them. Furthermore, in order to follow the corrections operated in the paper, a version of the manuscript having all the changes operated tracked (with the option track changes) has been also uploaded together with the last form of the manuscript (without the changes tracked).

The specific corrections operated according to the suggestions of the reviewers are given next together with detailed explanations.

 Reviewer # 2:

I have found the manuscript interesting. I think that the title is appropriate. The authors focus on study the mass tourist activity in Constanta through several conditions: social, economic, and environmental. The authors also make an effort to build a model and create a projection of the activities by 2050. These are the aims of the paper, which are clearly outlined both in the abstract and in the introduction section.

  • Thank you for your considerations about our study

The abstract fulfills its function of summarizing the paper’s content, and it remarks on the aims of the study. I would suggest adding the key findings. 

  • We added the following paragraph in the Abstract chapter

To create a model for coastal areas, the data used in this research must be accurate and consistent. Furthermore, correctly identifying indicators and their relationships is a critical step in conducting a thorough study. Last but not least, finding the calculation coefficient for the activity in question is critical, as collecting data from various activities might be challenging when trying to find a feasible model.

I would also propose to improve the readability of the text, using the same terms (p.e. PESTEL or PESTLE), and defining the acronyms, abbreviations, or initialisms the first time they appear in the text (p.e. INS: I suppose it refers to National Statistical Institute).

  • We have made the changes identified in the paper

There is repeated text in lines 99 to 103. 

  • We have made the changes identified in the paper

Regarding table 1, I suggest expressing it in thousands or millions,  indicating the equivalence in euros or dollars. 

  • We added the USD-RON exchange rate

The conclusions section could be extended a bit more, and I miss a discussion section. In addition, the limitations of the study should be indicated, such as the fact that it is making thirty-year predictions with data from only five periods.

  • In the Conclusions chapter, we have completed with the following paragraphs:
    • The PESTEL methodology was used to apply the multicriteria model to mass tourism in ConstanÈ›a. The model's use demonstrated the impact of relevant actions on three primary factors: economic, social, and environmental. By recognizing existing activities and determining the relationships between these activities, such a model can be applied to any tourist or seaside location.

Multicriteria analysis must be performed throughout time in order to detect probable changes in components and provide more accurate model predictions.Only considering the recommended indicators will not result in a long-term tourism activity. It's crucial to figure out which variables exert more or less pressure on the others. The supportability index must be determined for this. The equilibrium point must be determined where one activity's pressure must cease before the other activity vanishes.

  • One of the study's limitations was the data's availability and accuracy. Although most data is collected by public entities, it is not made available to the broader public. Furthermore, data gathering methods differ, and their accuracy in many circumstances is low. Another noteworthy limitation was the number of tourists visiting the area during the summer, despite the fact that the area is well-known for the fact that many local people offer their own houses for accommodation, but there are no relevant statistics on the number of tourists staying in these facilities. Another limitation of this study is that the activities in the coastal area are far more numerous and complex. As a result, in order to examine an effective situation, the interaction between these activities must be discovered, as well as the tolerance threshold between them. Last but not least, the study is time-limited in the sense that this analysis must be refreshed after a given period to determine whether the data entered is still relevant.

In general, the work seems interesting to me. The authors make a real contribution to the scientific literature with the analysis of the mass tourist activity.

I wish the authors good luck in revising their paper.

Thank you for your appreciations.
